# The Anti-Inflammatory Peptide *Tn*P Is a Candidate Molecule for Asthma Treatment

**DOI:** 10.3390/cells12060924

**Published:** 2023-03-17

**Authors:** Carla Lima, Maria Alice Pimentel Falcão, Felipe Justiniano Pinto, Jefferson Thiago Gonçalves Bernardo, Monica Lopes-Ferreira

**Affiliations:** Immunoregulation Unit, Laboratory of Applied Toxinology (CeTICs/FAPESP), Butantan Institute, São Paulo 05503900, Brazil; mpimentelfalcao@gmail.com (M.A.P.F.); felipejustinianopinto@gmail.com (F.J.P.); jefferson.bernardo@butantan.gov.br (J.T.G.B.); monica.lopesferreira@butantan.gov.br (M.L.-F.)

**Keywords:** *Tn*P, peptide drug, inflammatory diseases, asthma, lung remodeling

## Abstract

Asthma is the most common chronic lung disease, with increasing morbidity and mortality worldwide. Accumulation of peribronchial leukocytes is the hallmark of asthma, in particular, eosinophils, which have been reported as the primary cell associated with the induction of airway hyperresponsiveness. Continued exacerbation and accumulation of other leukocytes, such as neutrophils, Th1, and Th17 cells correlate with many of the long-term effects of asthma, such as airway remodeling. We have patented the *Tn*P family of synthetic cyclic peptides, which is in the preclinical phase of developmental studies for chronic inflammatory diseases. The aim of this work was to investigate whether *Tn*P could show anti-inflammatory activity in a murine model of asthma that includes a mixed phenotype of eosinophilic and neutrophilic inflammation. For this, Balb/c mice, sensitized with OVA and exposed to 1% challenge with OVA aerosol, were submitted to prophylactic treatment, receiving *Tn*P at 0.3 mg/kg orally, 1 h before each challenge. We found that sensitized mice challenged with OVA and treated with *Tn*P showed no airway hyperreactivity or lung remodeling. *Tn*P acts systemically in secondary lymphoid organs and locally in the lung, inhibiting the production of Th2/Th17 cytokines. Furthermore, *Tn*P prevented the infiltration of eosinophils and neutrophils in the BAL and lung tissue, inhibited the production of IgE/IgG1, prevented hyperplasia of mucus-producing cells, and decreased the thickening and deposition of sub-epithelial collagen. Our results showed *Tn*P as a candidate molecule for the treatment of airway remodeling associated with inflammatory diseases, such as asthma.

## 1. Introduction

Asthma is the most common chronic lung disease, with increasing morbidity and mortality worldwide, influenced by both genetic and environmental factors. Accumulation of peribronchial leukocytes is the hallmark of asthma, in particular, eosinophils, which have been reported as the primary cell associated with the induction of bronchial mucosal injury and airway hyperresponsiveness (Ahr) to stressful stimuli [1]. Continued exacerbation and accumulation of other leukocytes, such as neutrophils, Th (T helper)1 and Th17 cells correlate with many of the long-term effects of asthma [2,3], such as peribronchial thickening and fibrosis, leading to airway remodeling. This process is orchestrated by crosstalk of different cell types within the airway wall and submucosa [4].

The Global Initiative for Asthma (GINA) has recognized demographic, clinical, and pathophysiological characteristics that are clustered into distinct asthma phenotypes: allergic asthma with early-onset eosinophilic pediatric subendotype; infection-induced asthma with preschool wheezer pediatric subendotype; viral-exacerbated asthma; allergic bronchopulmonary mycosis (ABPM); aspirin-sensitive asthma; airflow obstruction caused by obesity; premenstrual asthma; neutrophilic asthma; elite-athlete asthma with exercise-induced asthma subendotype; cross-country skiers’ asthma; steroid-insensitive eosinophilic asthma; late-onset hypereosinophilic asthma; and severe steroid-dependent asthma [5].

Several standardized treatments generalized for all patients have focused on mitigating airway inflammation, including oral or inhaled corticosteroids and long-acting bronchodilators and other biological agents [6]. However, characterizing asthmatics by endotype can optimize management and drive personalized therapy targeting specific immunological mechanisms and biomarkers of the underlying disease.

Peptides from bacteria, fungi, plants, and animals have been characterized as having better therapeutic properties than their human counterparts, including higher selectivity, potency and in vivo stability, attracting significant attention for use as therapeutic candidates for immunotherapy of immune-mediated chronic inflammatory diseases [7,8,9]. Cabri et al. [10] reviewed data on peptides in clinical trials over the last 3 years and identified 58 peptides in different clinical phases: 13 entered the Phase 1, 26 are in Phase 2, 15 are in Phase 3, while four are close to approval, with the new drug application (NDA) already submitted to regulatory authorities. However, they found no available peptide for the treatment of multiple sclerosis or asthma at these stages of pharmaceutical development.

In this context, our group has been working with the *Tn*P family of synthetic cyclic peptides, first found in the venom of the Brazilian fish *Thalassophryne nattereri* [11], in preclinical models of autoimmune and inflammatory diseases. Initially we investigated its anti-inflammatory potential in a well-characterized mouse model for multiple sclerosis (MS), MOG-induced experimental autoimmune encephalomyelitis (EAE), and found that *Tn*P ameliorated the severity of the clinical signs of EAE, accompanied by the inhibition of neuroinflammation and improvement of the remyelination [12].

Among the multiple immunomodulatory effects of *Tn*P, it is interesting to note its action in the regulation of the entry of inflammatory cells, such as macrophages, Th1, and Th17 lymphocytes, into the central nervous system (CNS), inhibiting the expression of α4β1 integrin receptor in the blood-brain barrier that precedes perivascular infiltration and disease onset [12,13]. Furthermore, we demonstrated that EAE mice treated prophylactically with subcutaneously or orally applied *Tn*P exhibited protection against the development of CNS symptoms and inflammation similar to betaseron or fingolimod, but more efficiently than both in the long-term control of neuronal degeneration. Compared to glatiramer, *Tn*P was more efficient in switching off pathogenic cells as macrophages and microglia, and in activating regulatory cells [14].

In addition to the proven therapeutic effects of *Tn*P in a preclinical mouse model, analyses of its possible mechanisms of action [15,16], therapeutic index, safety and toxicology [17] were carried out in the zebrafish model organism.

To ensure a robust preclinical dossier essential for success from the early stages throughout clinical development, our next step was to investigate the ability of oral treatment with *Tn*P to control symptoms and the underlying immune response of asthma, using a murine model that includes a mixed phenotype of eosinophilic and neutrophilic inflammation. For this, we used an established model of OVA-induced acute asthma to analyze the ability of *Tn*P at 0.3 mg/kg to improve several aspects of the disease, including the recruitment of neutrophils and eosinophils to the bronchoalveolar lavage fluid (BAL), the release of inflammatory cytokines, and the production of anaphylactic antibodies. We also investigated the effect of *Tn*P on tissue remodeling and lung function.

## 2. Materials and Methods

### 2.1. Mice

Female 7–8-week-old BALB/c wild type mice were obtained from a colony at the Butantan Institute. Mice were maintained in sterile microisolators with sterile rodent feed and acidified water, and were housed in positive-pressure, air-conditioned units (25 °C, 50% relative humidity), on a 12 h light/dark cycle. The experiments were carried out under the laws of the National Council for Animal Experiment Control (CONCEA) and approved by the Butantan Institute’s Animal Use Ethics Commission (Permit Number: 1204/14).

### 2.2. OVA-induced Mouse Model of Asthma and TnP Treatment

On the first day of the experiments (day 0), mice were immunized with an injection of 10 μg OVA (Grade V, A-5503, Sigma Chem. Co., St Louis, MO, USA) into the s.c. tissue of the dorsal region. Starting on day 14, they were exposed to aerosolized OVA (1%, Grade V) for 20 min, for three days (asthma group). Mice injected only with sterile saline and challenged with PBS were considered as control (control group). During exposures, the animals were held in wire flow-through cage racks, and filtered air was drawn through the 0.5 m^3^ exposure chamber at a rate of 40 L/minute. Independent groups of mice were also treated 1 h before each aerosol challenge with dexamethasone (Dexa, D1756, Sigma) or *Tn*P trifluoroacetate compound (C_63_H_114_N_22_O_13_S_4_, H-Ile-Pro-Arg-Cys-Arg-Lys-Met-Pro-Gly-Val-Lys-Met-Cys-NH2, #P13821401, GenScript, purity 97.3%), both at 0.3 mg/kg. The animals were anesthetized with 0.2 mL Saffan (alphaxolone, 0.9% wt/vol plus alphadalone, 0.3% wt/vol; Vet Drug Ltd., Dunnington, York, UK) intraperitoneally, and 10 μL of saline alone (sham controls), or a solution of Dexa or *Tn*P in saline was inserted into the trachea by a nonsurgical technique. Samples were collected 24 h after the last allergen challenge.

### 2.3. Bronchoalveolar Lavage Fluid (BAL) and Cellular Analysis

BAL fluids were prepared by washing the lungs twice with 1 mL of PBS containing 10 mM EDTA. The cells were pooled by centrifugation at 400× *g* for 10 min at 4 °C. The supernatant (cell-free BAL fluid) was stored at −20 °C for cytokine analysis. Total cell counts were obtained using a hemocytometer. Differential cell counts based on morphological criteria were performed on cytospin preparations after staining with Diff-Quick (Protocol Hema 3, Fisher Healthcare Inc, Houston, TX, USA). For differential cell counts made under oil immersion microscopy at x 100 magnification, 300 leukocytes were enumerated and identified as mononuclear cells, eosinophils or neutrophils, based on staining and morphologic characteristics, using a Zeiss Axiophot microscope (Zeiss, Oberkochen, Germany). After BAL collection, the lungs, draining lymph nodes (DLN), and spleen were then removed. Tissues were minced and incubated with stirring at 37 °C for 30 min in HBSS with 1.3 mM EDTA, followed by treatment at 37 °C for 1 h with collagenase (150 U/mL; Invitrogen Life Technologies, Grand Island, NY, USA) in RPMI with 10% fetal bovine serum (FBS). The resulting cell suspension was washed three times, pelleted by centrifugation, and both the cell pellet and supernatant were used.

### 2.4. Determination of Airway Hyperresponsiveness (Ahr)

Airway function, in response to inhaled methacholine -MCh in a provocative concentration of 50 mg/Kg (causing a 300% increase in the baseline-enhanced pause (PenH) PC300), was assessed in conscious, spontaneously breathing animals, using a whole-body plethysmography system (Buxco Electronics, Petersfield, UK).

### 2.5. Titration of OVA-Specific Abs and Total IgE by ELISA

Blood was obtained by the puncture of the right ventricle of mice. Plasma was tested for IgG1 or IgG2a antibodies, using OVA-coated 96-well plates and biotinylated goat anti-mouse IgG1 or IgG2a antiserum. The reactions were developed with a streptavidin–horseradish peroxidase complex (Sigma), O-phenylenediamine (OPD) and H_2_O_2_, and the plates were read at 490 nm on an automated ELISA reader (Spectramax, Molecular Devices, San Jose, CA, USA). An IgE-specific ELISA was used to quantify total IgE antibody levels in the plasma, using matched Ab pairs (553413, Pharmingen and 1130-08, Southern Biotech, Birmingham, AL, USA), according to the manufacturer’s instructions. Samples were quantified by comparison with a standard curve.

### 2.6. Quantification of Cytokines and Chemokines

Cytokines and chemokines were measured in BAL, DNL and spleen by a specific two-site sandwich ELISA. The binding of biotinylated monoclonal antibodies was detected using a streptoavidin–horseradish peroxidase complex and TMB (3, 3’, 5, 5’-tetramethylbenzidine) substrate solution containing hydrogen peroxide. Detection limits were 7.8 pg/mL for IL-4 and IL-17A; 15.6 pg/mL for IL-5; 31.3 pg/mL for IL-10 and IFN-γ; 0.31 pg/mL for IL-13; and 500 pg/mL for Eotaxin (CCL11). Concentrations of IL-1β, IL-6, and TNF-α were determined by cytometric bead array (CBA), according to the manufacturer’s protocol (560485, BD Biosciences, San Jose, CA, USA), with minor modifications. The detection limits for IL-1β, IL-6, and TNF-α were 7.8 pg/mL, 1.4 pg/mL, and 0.9 pg/mL, respectively.

### 2.7. Histology of the Lungs

After BAL collection, the lungs were then removed and washed once with ice-cold HBSS, then fixed (10% formaldehyde) and paraffin-embedded. Paraffin-embedded sections were stained with haematoxilin/eosin (H & E), periodic acid-Schiff (PAS), and Picrosirius (PS). All slides were examined with light microscopy at a magnification of ×40 (Axio Imager A1, Carl Zeiss, Germany), and calibrated with a reference micrometer slide. For each group of six mice, four stained lung sections from each mouse were analyzed.

### 2.8. Statistical Analysis

All values were expressed as the mean ± SEM. Experiments using 3 to 5 mice per group were performed independently two times. Parametric data were evaluated using analysis of variance, followed by the Bonferroni test for multiple comparisons. Non-parametric data were assessed using the Mann–Whitney test. Differences were considered statistically significant at *p* < 0.05, using GraphPad Prism (Graph Pad Software, v6.02, 2013, La Jolla, CA, USA).

## 3. Results

### 3.1. Effects of Oral Administration of TnP on Inflammatory Cell Recruitment in BAL and Ahr

Eosinophils and neutrophils are one of the most numerous immune cells in the lungs of asthmatics, which have the potential to cause inflammation and tissue damage. Eosinophils and neutrophils may also modify their immediate environment by releasing mediators that correlate with airway, and parenchymal remodeling and airway hyperresponsiveness (Ahr), one of the defining symptoms of allergic asthma [18,19,20,21,22].

In this work, to investigate the ability of *Tn*P to modulate the recruitment of both cell types the lungs, we used an established murine model of OVA-induced acute asthma presenting a mixed phenotype of eosinophilic and neutrophilic inflammation, and compared it with dexamethasone, currently applied in the therapy of mild-to-moderate asthma [23,24,25].

The results in Figure 1 show that the number of total cells (Figure 1A) and eosinophils, which constituted ~50% of total BAL cells (Figure 1B), increased significantly in the airways of the asthma group compared to the mice in the control group that showed no inflammatory cells in the BAL. Dexa reduced the recruitment of total cells to the BAL by 69%, while *Tn*P reduced it by 64%, but both treatments completely abolished eosinophil recruitment (100%) to the BAL.

We have also examined the neutrophil population in the airways and found that successive OVA challenges of sensitized animals induce the influx of neutrophils to the BAL (10% of the total cells), and that both *Tn*P (75%) and Dexa (90%) treatments significantly decreased the neutrophil accumulation in the BAL compared to asthma-group mice (Figure 1C).

Next, Ahr was assessed by measuring the enhanced pause (PenH) using whole-body plethysmography, in response to inhaled MCh in conscious mice. The results shown in Figure 1D indicate that the PenH value of OVA-challenged mice was higher than that of the control group. However, a significant reduction of PenH values to the control level was observed in mice treated with *Tn*P or Dexa.

### 3.2. TnP Treatment Attenuates Airway Remodeling Induced by the OVA Challenge

Airway remodeling is an important feature of severe asthma and contributes to lung function reduction and obstruction of airflow. The permanent changes are characterized by an increased number of mucus-secreting goblet cells in the airway epithelium, with excessive mucus production, sub-epithelial fibrosis, and airway smooth muscle (ASM) hyperplasia/hypertrophy, and angiogenesis [26].

Next, we investigated the effect of *Tn*P on lung remodeling by assessing goblet cell hyperplasia, thickening, and sub-epithelial collagen deposition. In Figure 2 (left) we observed that H & E-stained lungs of the control mice exhibited normal lung tissue structure, including multiple alveoli with alveolar sacs, with ciliated columnar epithelium and a normal layer of smooth muscle fibers. Any influx of leukocytes, either around alveoli or into the lung interstitium, was observed (Figure 2A). In contrast, we observed that most alveolar sacs were collapsed, accompanied by an intense infiltrate of inflammatory cells at perivascular, peribronchiolar and interstitial areas in the lungs of the asthma group (Figure 2B) compared to the control mice, and the majority of cells infiltrating the lung were eosinophils followed by neutrophils and macrophages. The opposite was observed in the lungs of the *Tn*P-group mice (Figure 2D), which showed no eosinophil infiltration or other inflammatory cell types in the lungs. The treatment with Dexa (Figure 2C) decreased the recruitment of leukocytes in relation to the asthma group, but was not able to abolish it. An infiltration of peribronchiolar leukocytes was also seen.

In Figure 2 (right) we confirmed a substantial increase in the number of hyperplasic goblet cells that were PAS positive (intense pink) after OVA challenges in the asthma group (Figure 2F) compared to the control group (Figure 2E). Mice of the *Tn*P group (Figure 2H) showed few PAS-positive hyperplasic goblet cells in the airway epithelium compared to Dexa treatment, which maintained a higher number of hyperplastic goblet cells (Figure 2G).

Sub-epithelial matrix component deposition and thickening are characteristic of the remodeled airway. Anatomical details of collagen deposition and sub-epithelial thickening were provided by Picrosirius staining of the lung sections. OVA challenge of sensitized mice (Figure 3B) resulted in increased collagen deposition in the peribronchiolar region, with a sub-epithelial thickening of 12.13 μm (Figure 3F) compared to the control group (Figure 3A), which revealed minimal fibrous tissue deposition or thickening (3.69 μm, (Figure 3E). Additionally, we observed that treatment with *Tn*P significantly reduced both collagen deposition (Figure 3D) and thickening (6.8 μm, Figure 3H), but treatment with Dexa did not alter collagen deposition (Figure 3C) or sub-epithelial thickening (11.24 μm, Figure 3G).

### 3.3. TnP Treatment Inhibits Specific Anaphylactic Antibody Production

IL-4 and IL-13 drive allergen-specific IgE antibody production (Abs) in humans (and IgE and IgG1 in mice). The symptoms of the acute phase of asthma are dependent on mediators released after the degranulation of mast cells, activated by the cross-linking of FcεRI-bound IgE with antigen [27].

Repeated OVA challenge to sensitized mice (asthma group) induced significantly higher plasmatic levels of anaphylactic antibodies (Abs), such as total IgE (Figure 4A) and OVA-specific IgG1 (Figure 4B), as well as OVA-specific IgG2a (Figure 4C), when compared to the control-group mice. In contrast to the asthma group, *Tn*P-group mice presented significantly decreased levels of anaphylactic Abs (56% decrease of total IgE and 38% for IgG1), as well as IgG2a (71%). Interestingly, Dexa treatment of sensitized mice only had an effect on reducing IgG1 levels by 29%, but not total IgE or IgG2a levels, which remained similar to those of the asthma group.

### 3.4. TnP Suppresses OVA-Induced Th2-Associated Cytokine Production in BAL

Multiple cytokines, chemokines, and growth factors, released from both inflammatory and structural cells in the airway tissue, create a complex signaling environment that drives these structural changes [28]. Then, the modulation of factors driving remodeling is critical to alleviate the asthma symptoms and airway obstruction.

The next step was to evaluate the ability of *Tn*P to control the master effector cytokines of asthma. The measurement of these mediators in BAL, DLN, and spleen shows that the OVA challenge significantly increased the production of Th2 cytokines, including IL-4 (Figure 5A–C), IL-5 (Figure 5D), IL-13 (Figure 6A), and the chemokine eotaxin (Ccl11, Figure 6B).

In our model, we also observed the production of epithelial-derived innate cytokines, that in addition to Th2 cytokines, make a great contribution to the inflammatory response in asthma, such as IL-1β (Figure 7A), IL-6 (Figure 7B), and TNF-α (Figure 7C), the latter being considered a marker of neutrophilic asthma [29].

Furthermore, the asthma group showed increased production of IL-17A (Figure 7D) and IFN-γ (Figure 6C). Notably, *Tn*P oral treatment inhibited the production of cytokines associated with inflammatory leukocyte infiltration in the airways, Ahr and lung remodeling, such as IL-4, in all compartments: decrease of 58% in BAL, 97% in DLN, and 47% in the spleen (Figure 5A–C). IL-5 was diminished by 51% (Figure 5D), IL-1β by 89% (Figure 7A), IL-6 by 35% (Figure 7B), and TNF-α by 84% (Figure 7C) in the BAL of treated mice. A decrease of 58% in IL-17A production was also observed in the spleen (Figure 7D). However, our results in Figure 6 show that the treatment with oral *Tn*P was not able to reduce IL-13, eotaxin or IFN-γ production.

When mice were treated with Dexa, a reduction in the levels of IL-4 (58% in BAL and 97% in DLN), 75% of reduction of IL-5 (Figure 5D), and 50% of reduction of IL-13 in BAL (Figure 7A) was observed, although this treatment was not effective in reducing IL-4 in the spleen (Figure 5C), IL-1β, IL-6, IL-17, and TNF-α BAL levels (Figure 7), or the production of eotaxin (Figure 6B) or IFN-γ levels (Figure 6C).

## 4. Discussion

Asthma is a chronic inflammatory condition characterized by progressive structural changes in the bronchi, which can lead to significant and long-term impairment of the lung function. The therapeutic arsenal available today for the treatment of asthma, based on the use of corticosteroids combined with long-acting beta agonists, also includes leukotriene modifier, theophyllines, long-acting muscarinic agents, macrolide antibiotics, vitamin D, allergen immunotherapy, aspirin desensitization, dihydrofolate reductase inhibitor, hormones, CRTh2 antagonist, and monoclonal antibodies (mAbs), such as anti-IgE, anti-IL-4R, anti-IL-5, anti-IL-1R, anti-IL-13, anti-IL-17, anti-IL-25, anti-IL-33, and anti-TSLP [5]. However, these current therapeutic interventions that focus primarily on resolving inflammation do not translate into complete improvement of lung function deteriorated by airway remodeling.

A recent review by Cabri et al. [10] describes nine out of a total of 58 peptides in Phase 2 or 3 ongoing clinical trials for chronic inflammatory and autoimmune diseases, including: systemic lupus erythematosus, systemic inflammatory response syndrome, chronic kidney-disease-associated pruritus, acute respiratory distress syndrome, celiac disease adjuvant therapy, hearing and ocular inflammations, and inflammation of bone tissue. However, they found unavailable peptide for the treatment of multiple sclerosis or asthma, both which are chronic inflammatory diseases that present progressive forms associated with increased mortality, reduced quality of life and increased health costs [30,31].

Therefore, the identification of compounds and the development of therapeutic agents that incorporate anti-inflammatory functions and the ability to restore homeostatic functions (in the case of MS, remyelination and in the case of asthma, to prevent lung remodeling) are required.

In this field, bioactive peptides from marine organisms have gained increasing attention since the approval of the FDA in 2004 of the analgesic Ziconotide (Prialt^®^), found in marine snails [32]. Proof of the concept of *Tn*P in a preclinical model of EAE demonstrated its effectiveness in controlling clinical signs, inhibiting neuroinflammation and improving remyelination [12].

In this study, using a murine model of asthma with a mixed eosinophilic and neutrophilic phenotype, we investigated the ability of oral *Tn*P treatment to control the clinical symptom (Ahr), but also certain features of airway remodeling and key infiltration factors of leukocytes in the lung, and we tried to establish a correlation with the medication currently applied in the therapy of mild-to-moderate asthma.

First, we demonstrated that the presence of an eosinophilic infiltrate in the interstitial, perivascular and peribronchiolar regions of the lung and high IgE/IgG1 levels may favor increased Ahr and the initiation of lung remodeling, leading to the production of pro-fibrotic Th2 cytokines or other mediators responsible for the greater deposition of ECM components, such as collagen. Furthermore, the concurrent neutrophilic infiltration of the lungs associated with interconnected mechanisms, such as IL-17A, IFN-γ and IgG2a, indicates that neutrophil-rich inflammation was also driven by Th1/Th17 cells [28,33], relevant in the context of remodeling.

With this established model, we provide new evidence that *Tn*P has an anti-inflammatory action, improving Ahr and preserving lung function, accompanied by concomitant control of pro-inflammatory cytokine and anaphylactic antibody production, and eosinophilic and neutrophilic influx to the BAL, and mainly into parenchyma.

A key strength of the data presented in this study is that *Tn*P acts by inhibiting interstitial leukocyte infiltration, goblet cell hyperplasia, as well as fibrosis and sub-epithelial collagen deposition.

Although Dexa effectively controlled Ahr and the infiltration of the mixed-cell population to the BAL, it was not effective in preventing the interstitial and peribronchial infiltration of leukocytes. Moreover, Dexa did not control the production of innate cytokines (such as IL-1β, IL-6, TNF-α and IL-17A), IgE and mainly some aspects of tissue remodeling. This shows that oral treatment with Dexa fails to have a localized effect in the lungs.

Ahr depends on abnormal contraction of the airway smooth muscle cells (ASMCs) and the imbalance in the immune response. Of note, IL-6 not only induces the proliferation of ASMCs, but also promotes Th17 cell differentiation to increase Ahr. Likewise, IL-1β and TNF-α play prominent roles in increasing airway responsiveness, increasing airway inflammation and inducing imbalance in the immune response [34]. Then, the modulation of factors driving remodeling is critical to alleviate the asthma symptoms, Ahr, and airway obstruction and airflow limitation [35,36,37].

Nevertheless, there are some limitations in our study that must be considered. One limitation was the use of the OVA-induced acute asthma model (adjuvant free), which could be circumvented by the use of other preclinical models, which include inhaled and continuous exposure to house dust mite extract (HDM) for periods of up to 12 weeks, or induce structural changes in distal airways, or inflammation-independent remodeling models to demonstrate the effect of *Tn*P on epithelial and muscle cells and fibroblasts. An intense correlation between IL-25, IL-33, and thymic stromal lymphopoietin, the main cytokines derived from the epithelium, with the pathobiological responses, induced by aeroallergens, was observed in the airways, so the action of *Tn*P on these alarmins can be investigated.

*Tn*P studies using preclinical models could still be paired with studies involving clinical samples, such as those conducted with primary epithelial cells, fibroblasts or airway smooth muscle cells derived from the bronchoscopy of patients. These types of translational studies will provide a more comprehensive understanding of *Tn*P anti-inflammatory actions on the complex interactions of inflammation, bronchoconstriction, Ahr, and remodeling of asthma.

In conclusion, the results of *Tn*P compared to those of Dexa, which provide external validation of our results, show us that remodeling control can be achieved by therapeutic agents capable of acting locally, blocking chemotaxis and anchorage of cells in the interstitial matrix involved in the aggravation of the lung tissue damage. *Tn*P prevented the structural changes associated with the loss of function, which makes it a valid candidate molecule for the treatment of airway remodeling associated with chronic inflammatory diseases.

## Figures and Tables

**Figure 1 cells-12-00924-f001:**
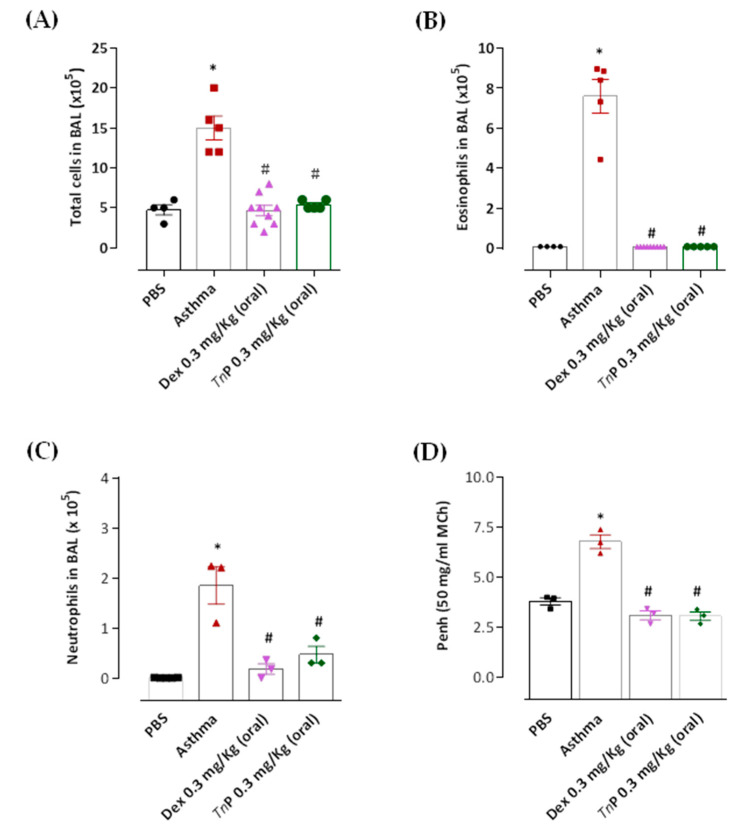
**Effect of *Tn*P on airway inflammation and *airway*
*hyperresponsiveness (Ahr)*.** Twenty-four hours after the last aerosol challenge, BAL from control, asthma, *Tn*P, or Dexa groups was collected for leukocyte (**A**), eosinophil (**B**), and neutrophil (**C**) counts. MCh responsiveness was assessed as changes in enhanced pause (Penh) in conscious animals using a whole-body plethysmography (**D**). The results represent the mean ± SEM of 3–5 animals/group. * *p* < 0.05 compared to the control group and # *p* < 0.05 compared to the asthma group.

**Figure 2 cells-12-00924-f002:**
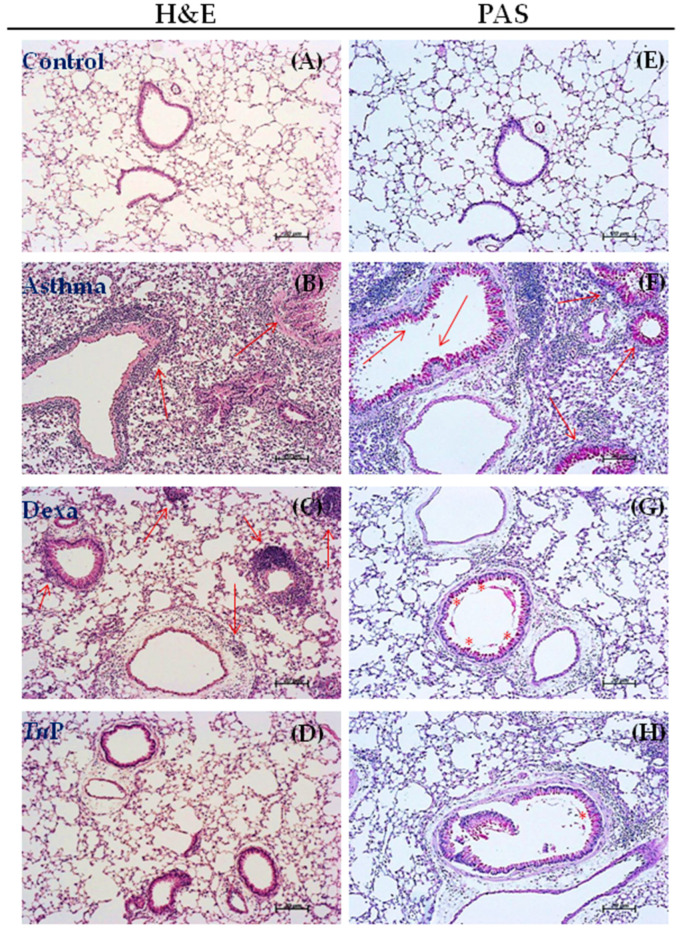
***Tn*P reduces lung cell infiltration and hyperplasia of mucus-producing cells.** Lung sections stained with H&E (**A**–**D**) and PAS (**E**–**H**) show normal cellularity or morphology in control mice (**A** or **E**), and perivascular, peribronchiolar and interstitial eosinophils in the asthma group, (**B**) with intense hyperplasic goblet cells (**F**). *Tn*P-group mice show no eosinophil infiltration (**D**) or hyperplasic goblet cells (**H**). Dexa decreased the recruitment of leukocytes in relation to the asthma group, but maintained infiltration of peribronchiolar leukocytes (**C**) and hyperplasic goblet cells (**G**). The red arrows indicate the intense perivascular, peribronchiolar and pulmonary interstitial leukocyte infiltration, and red asterisks are indicative of hyperplasic goblet cells.

**Figure 3 cells-12-00924-f003:**
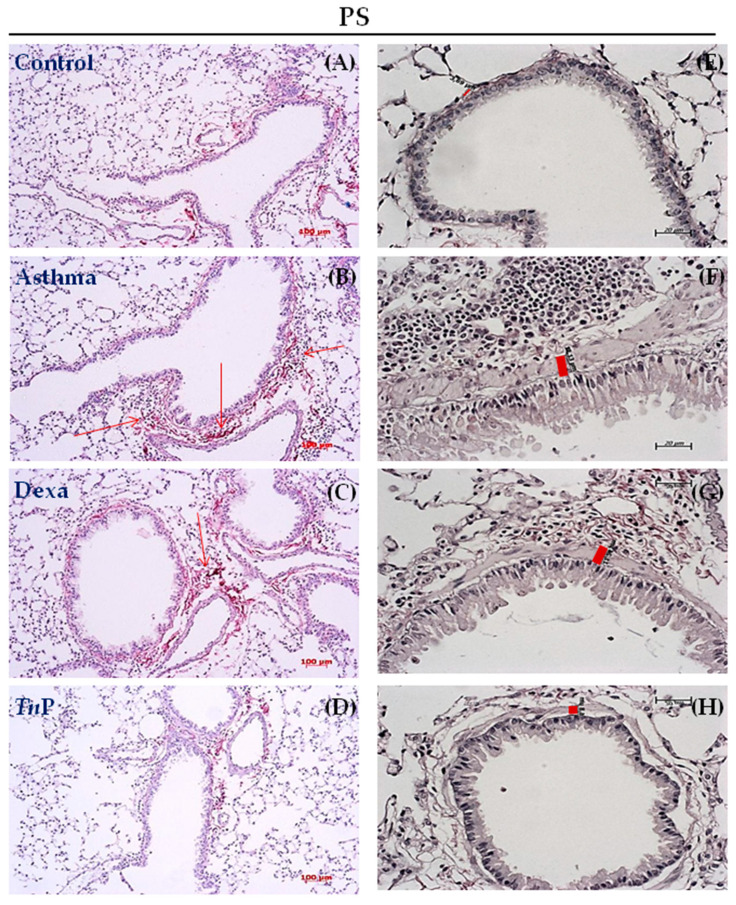
***Tn*P reduces airway remodeling.** Lung sections stained with PS (**A–H**) show the absence of thickness or sub-epithelial collagen deposit in control mice (**A,E**). Sub-epithelial fibrosis (**B,F**) was observed in the asthma group (**B,F**) and Dexa-treated mice (**C,G**), but not in the *Tn*P-treated mice (**D,H**). The red arrows indicate collagen fibers deposition and red bars indicate thickening.

**Figure 4 cells-12-00924-f004:**
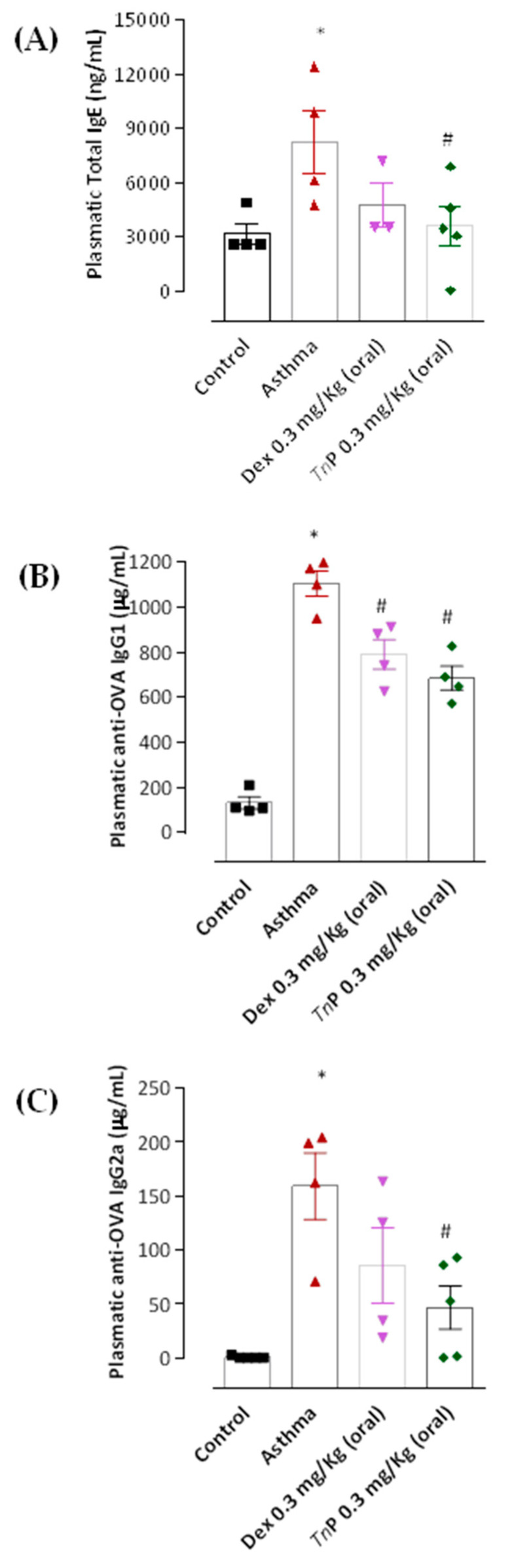
**Effect of *Tn*P on anaphylactic antibody titers.** Twenty-four hours after the last aerosol challenge, mice from the control, asthma, *Tn*P, or Dexa groups were bled for total (**A**), specific-OVA IgG1 (**B**), and specific-OVA IgG2a (**C**) determination by ELISA. The results represent the mean ± SEM of 3–5 animals/group. *****
*p* < 0.05 compared to the control group and # *p* < 0.05 compared to the asthma group.

**Figure 5 cells-12-00924-f005:**
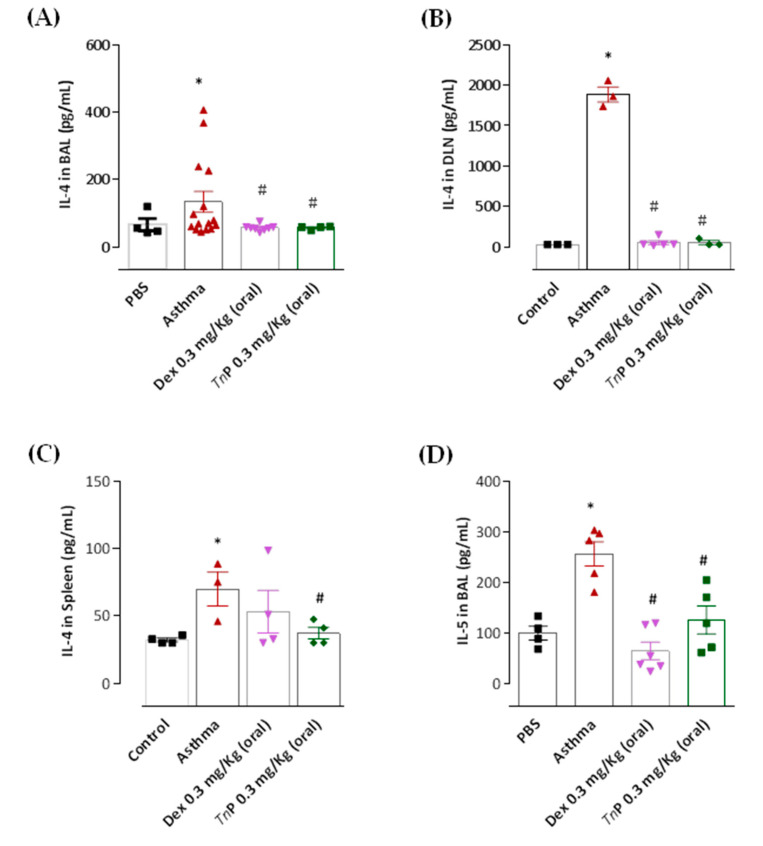
***Tn*P suppresses Th2 cytokine production.** OVA-sensitized mice subjected to challenge with aerosolized OVA, submitted or not to treatment with *Tn*P or Dexa, were killed after 24 h for analysis of IL-4 in BAL (**A**), draining lymph node (DLN, **B**), and spleen (**C**), and IL-5 in BAL (**D**) by cytokine-specific ELISA. *****
*p* < 0.05 compared to the control group and # *p* < 0.05 compared to the asthma group.

**Figure 6 cells-12-00924-f006:**
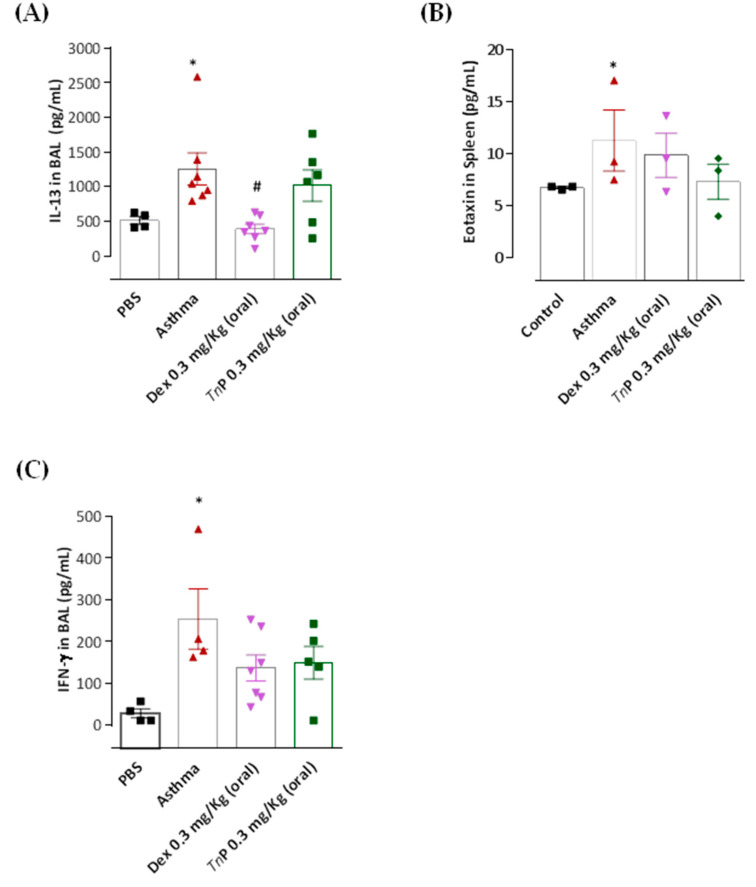
**Effect of *Tn*P on IL-13, eotaxin, and IFN-γ.** OVA-sensitized mice subjected to challenge with aerosolized OVA, submitted or not to treatment with *Tn*P or Dexa, were killed after 24 h for analysis of cytokines by ELISA. The treatment with oral *Tn*P was not able to reduce IL-13 (**A**), eotaxin (**B**) or IFN-γ (**C**) BAL production. *****
*p* < 0.05 compared to the control group and # *p* < 0.05 compared to the asthma group.

**Figure 7 cells-12-00924-f007:**
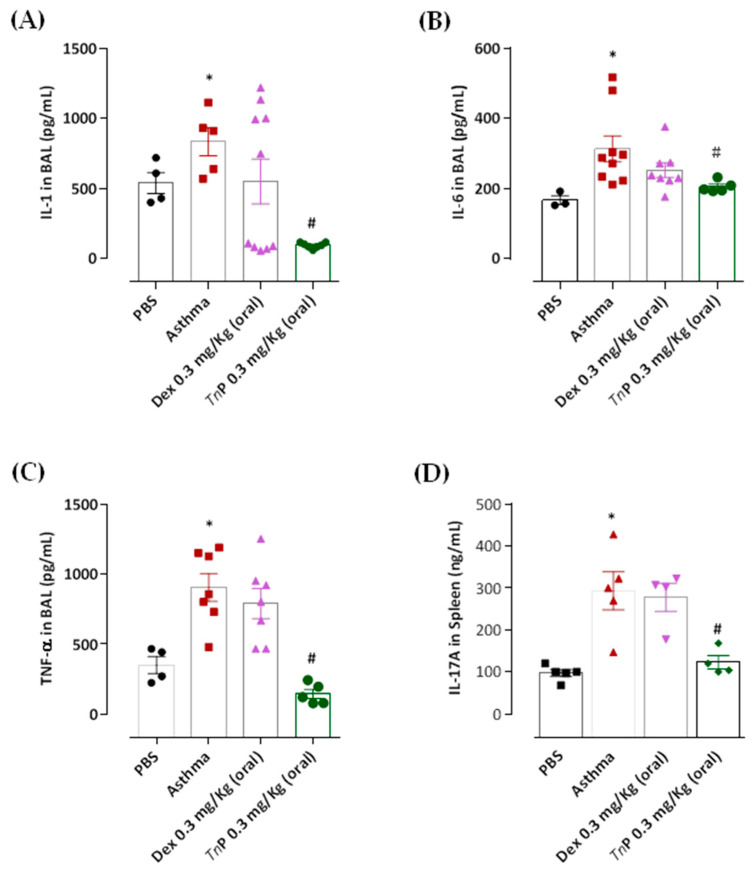
***Tn*P suppresses pro-inflammatory cytokine production.** OVA-sensitized mice subjected to challenge with aerosolized OVA, submitted or not to treatment with *Tn*P or Dexa, were killed after 24 h for analysis of IL-1β (**A**), IL-6 (**B**), TNF-α (**C**) in BAL and IL-17 in the spleen (**D**), by cytometric bead array. ******p* < 0.05 compared to the PBS control group and # *p* < 0.05 compared to the asthma group.

## Data Availability

Not applicable.

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
