# Peer review of "The Anti-Inflammatory Peptide TnP Is a Candidate Molecule for Asthma Treatment"

_cells, 2023, doi:10.3390/cells12060924_

Round 1

Reviewer 1 Report

he goal set by the authors is very ambitious. The prospect of a therapy for severe neutrophilic asthma, unresponsive to currently available drugs, is very suggestive. The authors illustrate in a precise and convincing way the research model, the statistical method used and the pathophysiological processes underlying their results. Finally, the results, albeit on animal models, are decidedly encouraging, since they demonstrate effective efficacy of the molecule used. Ultimately, the study could have a major impact on research and stimulate further studies on the efficacy of the therapy also on the human model.

Author Response

Response to reviewers

Reviewer #1

The goal set by the authors is very ambitious. The prospect of a therapy for severe neutrophilic asthma, unresponsive to currently available drugs, is very suggestive. The authors illustrate in a precise and convincing way the research model, the statistical method used and the pathophysiological processes underlying their results. Finally, the results, albeit on animal models, are decidedly encouraging, since they demonstrate effective efficacy of the molecule used. Ultimately, the study could have a major impact on research and stimulate further studies on the efficacy of the therapy also on the human model.

There are currently about 70 approved peptide drugs on the market, about 200 in clinical development and about 600 in the pre-clinical phase. The peptide market is growing twice as fast as the rest of the drug market, suggesting that peptides may occupy a larger specialty therapeutic niche soon. Among the highly successful peptide drugs on the market, which together represent about US$ 20 billion, we find Copaxone (glatiramer acetate), Lupron, Zoladex, Sandostatin and Velcade.

The objective of our research group is to test the anti-inflammatory capacity of molecules obtained from fish venoms, such as the TnP peptide using pre-clinical models of chronic inflammatory diseases. Indeed it is a great challenge, and we appreciate your positive opinion on the results presented here.

Reviewer 2 Report

The authors have done much work in this area of research. Well done. A few of my comments as below: 

The introduction focuses a lot on TnP in other diseases and on work already done (perhaps a little too much self-citation) and failed to establish the primary objectives / secondary objectives that their research is trying to prove adequately. Which phenotype / endotype of asthma do the authors intend to translate to the real world? Those not responding to glucocorticoids? If so, then their testing must be done on samples that do not respond to glucocorticoids? If the hypothesis is not set outright from the beginning, then what happens is the readers gets lost when the the authors are presenting the results (as with this article).

If your hypothesis is to add value to the asthma population not responding to glucocorticoids, how do you ensure the murine models are primed to achieve this objective? 

As mentioned, the results are rather confusing. On one hand, the authors mention some findings from their research but at some point, the reader realises that the authors are presenting findings from other research. In other words, the results and discussion have been combined together making it extremely confusing. The authors cite findings from previous studies but make it seem that it was found in this study. Very confusing. 

Please separate clearly your results and discussion into different sections. The results should only contain what has been discovered in this current study on mice and not from previous studies. Citations should be kept to the discussion section. 

The discussion section should then elaborate on ONE main point that was found in your study in each paragraph followed by supporting evidences / citations from work done and published prior. 

The article lacks a nice conclusion to tie in all the findings from the study. 

Author Response

Response to reviewers

Reviewer #2

The authors have done much work in this area of research. Well done. A few of my comments as below: 

  1. The introduction focuses a lot on TnP in other diseases and on work already done (perhaps a little too much self-citation) and failed to establish the primary objectives / secondary objectives that their research is trying to prove adequately. Which phenotype / endotype of asthma do the authors intend to translate to the real world? Those not responding to glucocorticoids? If so, then their testing must be done on samples that do not respond to glucocorticoids? If the hypothesis is not set outright from the beginning, then what happens is the readers gets lost when the the authors are presenting the results (as with this article).
  2. If your hypothesis is to add value to the asthma population not responding to glucocorticoids, how do you ensure the murine models are primed to achieve this objective?

We appreciate your time in evaluating the manuscript and consider your comments.

The Global Initiative for Asthma (GINA) has recognized demographic, clinical, and pathophysiological characteristics that are clustered into distinct asthma phenotypes: allergic asthma with early-onset eosinophilic pediatric subendotype; infection-induced asthma with preschool wheezer pediatric subendotype; viral-exacerbated asthma; allergic bronchopulmonary mycosis (ABPM); aspirin-sensitive asthma; airflow obstruction caused by obesity; premenstrual asthma; neutrophilic asthma; elite-athlete asthma with exercise-induced asthma subendotype; cross-country skiers’ asthma; steroid-insensitive eosinophilic asthma; late-onset hypereosinophilic asthma; and severe steroid-dependent asthma.

We corrected and amended the text throughout the entire manuscript to highlight the anti-inflammatory effect of TnP in asthma characterized by mixed, eosinophilic and neutrophilic inflammation.

Reddel, H.K.; Bacharier, L.B.; Bateman, E.D.; Brightling, C.E.; Brusselle, G.G.; Buhl, R.; Cruz, A.A.; Duijts, L.; Drazen, J.M.; FitzGerald, J.M.; et al. Global Initiative for Asthma Strategy 2021 Executive Summary and Rationale for Key Changes. Am. J. Respir. Crit. Care Med. 2022, 205, 17–35, doi:10.1164/rccm.202109-2205PP.

  1. As mentioned, the results are rather confusing. On one hand, the authors mention some findings from their research but at some point, the reader realises that the authors are presenting findings from other research. In other words, the results and discussion have been combined together making it extremely confusing. The authors cite findings from previous studies but make it seem that it was found in this study. Very confusing. 
  2. Please separate clearly your results and discussion into different sections. The results should only contain what has been discovered in this current study on mice and not from previous studies. Citations should be kept to the discussion section. 
  3. The discussion section should then elaborate on ONE main point that was found in your study in each paragraph followed by supporting evidences / citations from work done and published prior.

The Results and Discussion sections were separated to better expose the experimental evidence.

Following your suggestions, the discussion was rewritten to present the experimental evidence that shows the effectiveness of TnP in controlling some of the hallmarks of asthma including persistent inflammation and tissue remodeling, a mixed inflammatory phenotype, and a decline in lung function.

  1. The article lacks a nice conclusion to tie in all the findings from the study. 

After readjusting the writing of the manuscript, separating the results from the discussion, we elaborated a more appropriate conclusion, which highlights the relevance of the findings.

Reviewer 3 Report

1.     It is better to conduct trials against clinical drugs used in the treatment of asthma to compare and analyze the different mechanisms in the treatment of asthma and to elucidate the advantages and disadvantages.

2.     In addition, there are some grammar mistakes in manuscript, please edit the article by the native speaker.  

Author Response

Response to reviewers

Reviewer #3

  1. It is better to conduct trials against clinical drugs used in the treatment of asthma to compare and analyze the different mechanisms in the treatment of asthma and to elucidate the advantages and disadvantages.

We appreciate your time in evaluating the manuscript and consider your comments.

The Global Asthma Initiative (GINA) Strategy Report (2021) recommends for reduction of airway inflammation and lung dysfunction, control symptoms, and reduction of the risks of exacerbations, even in mild asthma of adults and adolescents β2- adrenergic agonists in combination with therapy containing inhaled corticosteroids: budesonide–formoterol or beclometasone–formoterol.

However, in several countries, dexamethasone has been applied to control severe or disabling allergic conditions, not susceptible to adequate attempts of conventional treatment, such as asthma, seasonal or perennial allergic rhinitis, contact dermatitis, atopic dermatitis, serum sickness, hypersensitivity reactions to medications.

Oral prednisone and dexamethasone are the currently recommended by Canadian Thoracic Society, Canadian Association of Emergency Physicians, British Thoracic Society, and American National Asthma Education and Prevention Program for moderate to severe exacerbations of asthma and for mild exacerbations not responsive to bronchodilator therapy.

Traditionally, mild-to-moderate pediatric asthma exacerbations have been treated with a short course of oral steroids—often 5 days of prednisone or prednisolone. Recent evidence suggests a similar outcome can be achieved with a single dose of dexamethasone, which has a longer half-life and powerful anti-inflammatory effects, along with easier administration and compliance. Single-dose dexamethasone offers a simple and reliable treatment for these patients in office, urgent care, and emergency department settings. Dexamethasone can be given for 1 to 5 days at a dose ranging from 0.3 to 0.6 mg/kg daily. Dexamethasone is a long-acting glucocorticoid with a half-life of 36 to 72 hours, and is 6 times more potent than prednisone. Prednisone is shorter acting, with a half-life of 18 to 36 hours.

In our work, we used dexamethasone to compare with TnP, once of its current application in the therapy of mild-to-moderate asthma and because its salt is available from Sigma (D1756).

Reddel HK, Bacharier LB, Bateman ED, Brightling CE, Brusselle GG, Buhl R, Cruz AA, Duijts L, Drazen JM, FitzGerald JM, Fleming LJ, Inoue H, Ko FW, Krishnan JA, Levy ML, Lin J, Mortimer K, Pitrez PM, Sheikh A, Yorgancioglu AA, Boulet LP. Global Initiative for Asthma Strategy 2021: Executive Summary and Rationale for Key Changes. Am J Respir Crit Care Med. 2022 Jan 1;205(1):17-35. doi: 10.1164/rccm.202109-2205PP.

Shefrin AE, Goldman RD. Use of dexamethasone and prednisone in acute asthma exacerbations in pediatric patients. Can Fam Physician. 2009 Jul;55(7):704-6.

Banoth, Bhaskar; Verma, Anjali; Bhalla, Kapil; Khanna, Alok; Holla, Saraswathi; Yadav, Swati. Comparative effectiveness of oral dexamethasone vs. oral prednisolone for acute exacerbation of asthma: A randomized control trial. Journal of Family Medicine and Primary Care 11(4):p 1395-1400, April 2022. | DOI: 10.4103/jfmpc.jfmpc_1210_21

Cross KP, Paul RI, Goldman RD. Single-dose dexamethasone for mild-to-moderate asthma exacerbations: effective, easy, and acceptable. Can Fam Physician. 2011 Oct;57(10):1134-6.

Kwah JH, Peters AT. Asthma in adults: Principles of treatment. Allergy Asthma Proc. 2019 Nov 1;40(6):396-402. doi: 10.2500/aap.2019.40.4256.

Chung KF. Clinical management of severe therapy-resistant asthma. Expert Rev Respir Med. 2017 May;11(5):395-402. doi: 10.1080/17476348.2017.1317597.

  1. In addition, there are some grammar mistakes in manuscript, please edit the article by the native speaker.

The manuscript was rechecked for writing in the English language to avoid grammatical or spelling errors.

Round 2

Reviewer 2 Report

No further comments 

Author Response

As suggested, we have made changes to the text of the main manuscript in the Discussion including the limitations of the study and its strength.

Reviewer 3 Report

The manuscript was adequately modified by the authors. I thought that it can be accepted in current form.

Author Response

(The authors gave the same response as above.)
